# Molecule Design by Latent Prompt Transformer

**Deqian Kong**
UCLA

**Yuhao Huang**
Xi'an Jiaotong Univerisity

**Jianwen Xie**
BioMap

**Ying Nian Wu**
UCLA

## Abstract

This paper proposes a latent prompt Transformer model for solving challenging optimization problems such as molecule design, where the goal is to find molecules with optimal values of a target chemical or biological property that can be computed by an existing software. Our proposed model consists of three components. (1) A latent vector whose prior distribution is modeled by a Unet transformation of a Gaussian white noise vector. (2) A molecule generation model that generates the string-based representation of molecule conditional on the latent vector in (1). We adopt the causal Transformer model that takes the latent vector in (1) as prompt. (3) A property prediction model that predicts the value of the target property of a molecule based on a non-linear regression on the latent vector in (1). We call the proposed model the latent prompt Transformer model. After initial training of the model on existing molecules and their property values, we then gradually shift the model distribution towards the region that supports desired values of the target property for the purpose of molecule design. Our experiments show that our proposed model achieves state of the art performances on several benchmark molecule design tasks.

## 1 Introduction

In drug discovery, identifying or designing molecules with specific pharmacological or chemical attributes, such as enhanced drug-likeness or high binding affinity to target proteins, is of paramount importance. However, navigating the vast space of potential drug-like molecules presents a daunting challenge.

To address this challenge, several contemporary research avenues have emerged. One prominent approach involves the application of latent space generative models. This approach strives to translate the discrete molecule graph into a more manageable continuous latent vector. Once translated, molecular properties can be optimized within the continuous latent space utilizing various strategies [Gómez-Bombarelli et al., 2018, Kusner et al., 2017, Jin et al., 2018, Eckmann et al., 2022, Kong et al., 2023]. Another avenue of research is more direct, employing reinforcement learning algorithms to fine-tune molecular attributes directly within the molecule graph space [You et al., 2018, De Cao and Kipf, 2018, Zhou et al., 2019, Shi et al., 2020, Luo et al., 2021]. Diverse alternative methodologies have also gained traction, such as genetic algorithms [Nigam et al., 2020], particle-swarm strategies [Winter et al., 2019] and scaffolding tree [Fu et al., 2021].

In this paper, we propose a novel latent prompt Transformer model for molecule design. As in most existing work on molecule design, we assume that the value of a property of interest of a given molecule can be obtained by querying an existing software such as RDKit [Landrum et al.] and AutoDock-GPU [Santos-Martins et al., 2021]. Thus in this paper, we solely focus on optimizing the value of the target property. In our paper, we work with string-based representation of molecules, such as the commonly used SMILES [Weininger, 1988] and the more recently proposed SELFIES [Krenn et al., 2020] and its variant [Cheng et al., 2023].

NeurIPS 2023 AI for Science Workshop.

Our proposed model belongs to the latent space generative modeling approach mentioned above. Our model consists of three components. (1) A latent vector whose prior distribution is modeled by a Unet transformation of a Gaussian white noise vector. (2) A molecule generation model that generates the string-based representation of molecule given the latent vector in (1). We adopt the causal Transformer model that takes the latent vector in (1) as prompt. (3) A property prediction model that predicts the value of the target property of a molecule based on a non-linear regression on the latent vector in (1). We call the proposed model the latent prompt Transformer model.

After initial training of the model on existing molecules and their property values, we then gradually shift the model distribution towards the region that supports desired values of the target property for the purpose of molecule design. Our experiments show that our proposed model achieves state of the art performances on several benchmark molecule design tasks.

The contributions of our paper are as follows. (1) We propose a novel latent prompt Transformer model for modeling the joint distribution of the molecules and their values of target property. (2) We develop the approximate maximum likelihood learning algorithm to fit the model to the training molecules and their properties. We also employ a gradual distribution shifting algorithm that shifts our model distribution towards the region that supports desired values of target property. (3) We conduct experiments on single-objective and multi-objective molecule design and our experiments achieve new state of the arts on various benchmark tasks.

## 2  Method

### 2.1  Latent Prompt Transformer

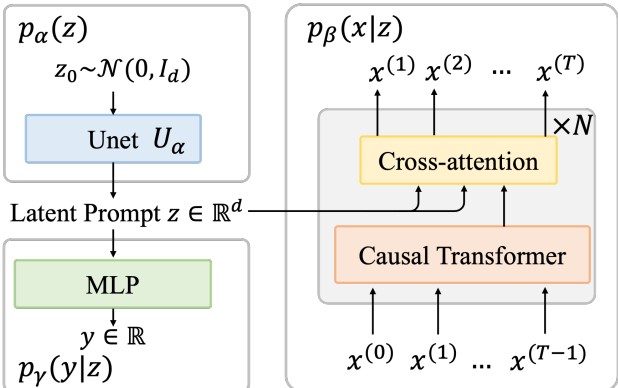

Figure 1: Latent Prompt Transformer. $x$ is the string-based representation of molecule. $y$ is the value of a target property. $z$ is the latent vector. $z_0 \sim \mathcal{N}(0, I_d)$. (1) The prior distribution of $z$ is modeled by a Unet transformation of $z_0$, i.e., $z = U_\alpha(z_0)$. Given $z$, $x$ and $y$ are independent. (2) $p_\beta(x|z)$ is the generation model, parametrized by a causal Transformer with $z$ serving as the prompt. (3) $p_\gamma(y|z)$ is the property prediction model, which is a non-linear regression on $z$ parametrized by a multi-layer perceptron (MLP).

Our model is illustrated by Fig. 1. Suppose $x = (x^{(1)}, ..., x^{(t)}, ..., x^{(T)})$ is a molecule string in SELFIES [Krenn et al., 2020], $y \in \mathbb{R}$ is the value of the target property of interest, and $z \in \mathbb{R}^d$ is the latent vector. We define the following model as the latent prompt Transformer (LPT):

$$z \sim p_\alpha(z), \quad [x|z] \sim p_\beta(x|z), \quad [y|z] \sim p_\gamma(y|z), \tag{1}$$

where $p_\alpha(z)$ is a prior model with parameters $\alpha$. $z$ serves as the latent prompt of the molecule generation model $p_\beta(x|z)$ parameterized by a causal Transformer with parameter $\beta$. $p_\gamma(y|z)$ is a property prediction model with parameter $\gamma$.

For the prior model, $p_\alpha(z)$ is formulated as a learnable neural transport from an uninformative prior,

$$z = U_\alpha(z_0), \tag{2}$$

where $z_0$ is assumed to be isotropic Gaussian $z_0 \sim \mathcal{N}(0, I_d)$, and $U_\alpha(\cdot)$ is parametrized by an expressive neural network such as a Unet with parameter $\alpha$.

The molecule generation model $p_\beta(x|z)$ is a conditional autoregressive model,

$$p_\beta(x|z) = \prod_{t=1}^{T} p_\beta(x^{(t)}|x^{(0)}, ..., x^{(t-1)}, z) \tag{3}$$

which is parameterized by a causal Transformer with parameter $\beta$. Note that the latent vector $z$ controls every step of the autoregressive generation and it functions as a soft prompt that controls the generation of molecules.

Given a molecule $x$, let $y$ denote the value of the target property, such as drug likeliness or protein binding affinity. One can determine the estimated value of this property using open-source software such as RDKit [Landrum et al.] and AutoDock-GPU [Santos-Martins et al., 2021].

Given $z$, we posit that $x$ and $y$ are conditionally independent. Under this assumption, LPT defines the joint distribution

$$p_\theta(x, y, z) = p_\alpha(z)p_\beta(x|z)p_\gamma(y|z), \tag{4}$$

where $\theta = (\alpha, \beta, \gamma)$. We use the marginal distribution $p_\theta(x, y) = \int p_\theta(x, y, z)dz$ to approximate the data distribution $p_{\text{data}}(x, y)$.

For the property prediction model, we assume

$$p_\gamma(y|z) = \frac{1}{\sqrt{2\pi\sigma^2}} \exp\left(-\frac{1}{2\sigma^2}(y - s_\gamma(z))^2\right), \tag{5}$$

where $s_\gamma(z)$ is a small multi-layer perceptron (MLP), predicting $y$ based on the latent prompt $z$. The variance $\sigma^2$ is treated as a hyper-parameter. Given this formulation, the latent prompt $z$ is aware of the property value while generating the molecule.

For tasks involving multi-objective design with target properties $\boldsymbol{y} = \{y_j\}_{j=1}^{M}$, the regression model can be extended to $p_\gamma(\boldsymbol{y}|z) = \prod_{j=1}^{M} p_{\gamma_j}(y_i|z)$, where each $p_{\gamma_j}(y_i|z)$ is parametrized as in (5). Without much loss of generality, we shall focus on the single-objective setting in the following sections.

## 2.2 Learning

Suppose we observe training examples $\{(x_i, y_i), i = 1, ..., n\}$. The log-likelihood function is $L(\theta) = \sum_{i=1}^{n} \log p_\theta(x_i, y_i)$.

Since $z = U_\alpha(z_0)$, we can also write the model as

$$p_\theta(x, y) = \int p_\beta(x|z = U_\alpha(z_0))p_\gamma(y|z = U_\alpha(z_0))p_0(z_0)dz_0, \tag{6}$$

where $p_0(z_0) \sim \mathcal{N}(0, I_d)$. The learning gradient can be calculated according to

$$\nabla_\theta \log p_\theta(x, y) = \mathbb{E}_{p_\theta(z_0|x,y)}\left[\nabla_\theta(\log p_\beta(x|U_\alpha(z_0)) + \log p_\gamma(y|U_\alpha(z_0)))\right]. \tag{7}$$

Given an example $(x, y)$, the learning gradient for the prior model is

$$\delta_\alpha(x, y) = \mathbb{E}_{p_\theta(z_0|x,y)}[\nabla_\alpha(\log p_\beta(x|U_\alpha(z_0)) + \log p_\gamma(y|U_\alpha(z_0)))]. \tag{8}$$

The learning gradient for the molecule generation Transformer is

$$\delta_\beta(x, y) = \mathbb{E}_{p_\theta(z_0|x,y)}[\nabla_\beta \log p_\beta(x|U_\alpha(z_0))]. \tag{9}$$

The learning gradient for the property regression model is

$$\delta_\gamma(x, y) = \mathbb{E}_{p_\theta(z_0|x,y)}[\nabla_\gamma \log p_\gamma(y|U_\alpha(z_0))]. \tag{10}$$

Estimating expectations in Eqs. (8) to (10) requires MCMC sampling of the posterior distribution $p_\theta(z_0|x, y)$. We recruit Langevin dynamics [Neal, 2011, Han et al., 2017]. For a target distribution $\pi(z)$, the dynamics iterates

$$z^{\tau+1} = z^\tau + s\nabla_z \log \pi(z^\tau) + \sqrt{2s}\epsilon^\tau, \tag{11}$$

where $\tau$ indexes the time step of the Langevin dynamics, $s$ is step size, and $\epsilon_\tau \sim \mathcal{N}(0, I_d)$ is the Gaussian white noise. $\pi(z)$ here is the posterior $p_\theta(z_0|x, y)$, and the gradient can be efficiently computed by back-propagation.

We initialize $z_0^{\tau=0} \sim \mathcal{N}(0, I_d)$, and employ $N$ steps of Langevin dynamics (e.g. $N = 15$) for approximate sampling from the posterior distribution, rendering our learning algorithm as an approximate maximum likelihood estimation. See [Pang et al., 2020, Nijkamp et al., 2020, Xie et al., 2023] for a theoretical understanding of the learning algorithm based on the finite-step MCMC.

In practical applications with multiple molecular generation tasks, with each characterized by a different target property $y$, each model $p_\theta(x, y)$ may necessitate separate training. For the sake of efficiency, we adopt a two-stage training approach. In the first stage, we train the model on molecules alone while ignoring the properties by maximizing $\log p_\theta(x) = \log \int p_\theta(x, z) dz$. In the second stage, we fine-tune the model for the specific target property under consideration using Eqs. (8) to (10). To be specific, for the first pre-training stage, the learning gradient is

$$\nabla_\theta \log p_\theta(x) = \mathbb{E}_{p_\theta(z_0|x)} \left[ \nabla_\beta \log p_\theta(x|U_\alpha(z_0)) \right], \tag{12}$$

so that for a training example $(x, y)$, the learning gradients for $\alpha$ and $\beta$ are

$$\delta_\alpha(x) = \mathbb{E}_{p_\theta(z_0|x)} [\nabla_\alpha \log p_\beta(x|U_\alpha(z_0))], \tag{13}$$

$$\delta_\beta(x) = \mathbb{E}_{p_\theta(z_0|x)} [\nabla_\beta \log p_\beta(x|U_\alpha(z_0))]. \tag{14}$$

The learning algorithms for pre-training LPT and fine-tuning LPT are summarized in Algorithms 1 and 2 respectively. This two-stage approach is also adaptable for semi-supervised scenarios where property values might be scarce.

---

**Algorithm 1:** Pre-training LPT solely on molecules

**input** : Learning iterations $T$, learning rates for the prior, generation models $\{\eta_0, \eta_1\}$, initial parameters $\theta_0 = (\alpha_0, \beta_0)$, observed examples $\{x_i\}_{i=1}^n$, batch size $m$, number of posterior sampling steps $N_0$, and posterior sampling step size $s_0$.

**output** : $\theta_T = (\alpha_T, \beta_T, \gamma_T)$.

**for** $t = 0 : T - 1$ **do**

 1. **Mini-batch**: Sample observed examples $\{x_i\}_{i=1}^m$.
 2. **Posterior sampling**: For each $x_i$, sample $z_0 \sim p_{\theta_t}(z_0|x_i)$ using Eq. (11), where the target distribution $\pi$ is $p_{\theta_t}(z_0|x_i)$, and $s = s_0$, $N = N_0$.
 3. **Update prior model**: $\alpha_{t+1} = \alpha_t + \eta_0 \frac{1}{m} \sum_{i=1}^m [\delta_\alpha(x_i)]$ as in Eq. (13).
 4. **Update generation model**: $\beta_{t+1} = \beta_t + \eta_1 \frac{1}{m} \sum_{i=1}^m [\delta_\beta(x_i)]$ as in Eq. (14).

---

### 2.3 Initial Training and Conditioned Generation

We can first pre-train the model on a dataset of existing molecules, such as ZINC [Irwin et al., 2012] using Algorithm 1 . Given a target property, we can then fine-tune the model using Algorithm 2, where for each molecule $x$ in the training dataset, we can obtain the corresponding $y$ by querying an existing software that estimates the value of the target property, such as RDKit [Landrum et al.] and AutoDock-GPU [Santos-Martins et al., 2021]. In this paper, we treat the values produced by the software as the ground-truth values.

Given a trained model, we can generate a molecule $x$ conditional on a given value $y$ of the target property by sampling from $p_\theta(x|y)$. The sampling can be accomplished by the following two steps. Step 1: sample $z \sim p_\theta(z|y)$, and Step 2: sample $x \sim p_\beta(x|z)$. To accomplish Step 1, we can first sample $z_0 \sim p_\theta(z_0|y) \propto p_0(z_0)p_\gamma(y \mid z = U_\alpha(z_0))$ by Langevin dynamics, and then let $z = U_\alpha(z_0)$.

### 2.4 Gradual Distribution Shifting

Given an initially trained model, for the purpose of molecule design, it is tempting to set $y$ at a desired value $y^*$, and then generate $x \sim p_\theta(x|y^*)$. The problem is that $y^*$ may be out of the range of

---

**Algorithm 2:** Fine-tuning or learning LPT on both molecules and their properties

---

**input** : Learning iterations $T$, learning rates for the prior, generation, and regression models $\{\eta_0, \eta_1, \eta_2\}$, initial parameters $\theta_0 = (\alpha_0, \beta_0, \gamma_0)$, observed samples $\{(x_i, y_i)\}_{i=1}^n$, batch size $m$, number of posterior sampling steps $N_1$, and posterior sampling step size $s_1$.

**output** : $\theta_T = (\alpha_T, \beta_T, \gamma_T)$.

**for** $t = 0 : T - 1$ **do**

    1. **Mini-batch**: Sample observed examples $\{(x_i, y_i)\}_{i=1}^m$.

    2. **Posterior sampling**: For each $(x_i, y_i)$, sample $z_0 \sim p_{\theta_t}(z_0|x_i, y_i)$ using Eq. (11), where the target distribution $\pi$ is $p_{\theta_t}(z_0|x_i, y_i)$, and $s = s_1$, $N = N_1$.

    3. **Update prior model**: $\alpha_{t+1} = \alpha_t + \eta_0 \frac{1}{m} \sum_{i=1}^m [\delta_\alpha(x_i, y_i)]$ as in Eq. (8).

    4. **Update generation model**: $\beta_{t+1} = \beta_t + \eta_1 \frac{1}{m} \sum_{i=1}^m [\delta_\beta(x_i, y_i)]$ as in Eq. (9).

    5. **Update regression model**: $\gamma_{t+1} = \gamma_t + \eta_2 \frac{1}{m} \sum_{i=1}^m [\delta_\gamma(x_i, y_i)]$ as in Eq. (10).

---

---

**Algorithm 3:** Sampling with Gradual Distribution Shifting (SGDS).

---

**input** : Shift iterations $T$, initial pre-trained parameters $\theta_0 = (\alpha_0, \beta_0, \gamma_0)$, initial samples $\mathcal{D}^0 = \{(x_i^0, y_i^0, z_i^0)\}_{i=1}^n$ from the data distribution boundary, shift magnitude $\Delta_y$, a score function $S(x)$ and $m$ generated samples in each iteration.

**output** : $\{(x_i^T, y_i^T)\}_{i=1}^n$.

**for** $t = 1 : T$ **do**

    1. **Dataset Creation**:
    Generate $\{z_i^{t+1}, x_i^{t+1}\}_{i=1}^m$ such that $z_i^{t+1} \sim p_{\theta_t}(z_0|y = y^t + \Delta_y)$ and $x_i^{t+1} \sim p_{\beta_t}(x|z^{t+1})$.
    Annotate $\{y_i^{t+1} = S(x_i^{t+1})\}_{i=1}^m$.
    Create $\mathcal{G}^{t+1} = \{x_i^{t+1}, y_i^{t+1}, z_i^{t+1}\}_{i=1}^m \cup \mathcal{D}^t = \{x_i^{t+1}, y_i^{t+1}, z_i^{t+1}\}_{i=1}^{n+m}$.
    2. **Model Shift**:
    Rank $\mathcal{G}^{t+1}$ based on target property $y^{t+1}$ yielding $\mathcal{G}^{t+1} = \{x_{(i)}^{t+1}, y_{(i)}^{t+1}, z_{(i)}^{t+1}\}_{i=1}^{n+m}$ where $y_{(1)}^{t+1} \geq y_{(2)}^{t+1} \geq \cdots \geq y_{(m+n)}^{t+1}$.
    Create $\mathcal{D}^{t+1} = \{x_{(i)}^{t+1}, y_{(i)}^{t+1}, z_{(i)}^{t+1}\}_{i=1}^n$.
    Update $\theta_{t+1} = \underset{\theta}{\arg\max} \, \mathbb{E}_{(x,y) \sim \mathcal{D}^{t+1}}[\log p_\theta(x, y)]$ using Algorithm 2.

---

the learned distribution $p_\theta(x, y, z)$ or more specifically its marginal distribution $p_\theta(y)$. As a result, sampling from $p_\theta(x|y^*)$ amounts to out-of-distribution (OOD) extrapolation, which can be unreliable.

The *sampling with gradual distribution shifting* (SGDS) algorithm [Kong et al., 2023] was proposed to address the above problem. In this algorithm, we can maintain a top-$n$ *shifting dataset* $\mathcal{D}^t = \{x_i^t, y_i^t, z_i^t\}_{i=1}^n$, where $t$ denotes the iteration in the SGDS algorithm. To initialize at $t = 0$, we obtain $\mathcal{D}^0$ by selecting the top-$n$ molecules from the initial training dataset such as ZINC [Irwin et al., 2012] by ranking them based on their values of target property. That is, $\mathcal{D}^0$ is selected at the boundary of the initial training set. In each iteration of SGDS, we incrementally increase the values of the properties in the top-$n$ shifting dataset, and then generate new molecules conditional on the increased values. Because the incrementally increased values are expected to be close to the current model distribution, the conditional generation is expected to be reliable. Nonetheless, we still query the software to obtain the ground-truth values of the target property for the newly generated molecules. We then update the top-$n$ shifting dataset by ranking the molecules in the current shifting dataset as well as the newly generated molecules based on the ground-truth values of the property. For this new top-$n$ shifting dataset, we then re-learn our model, in order for the model to catch up with the shifting data, so that further incremental shifting and generation can still be reliable. More specifically, each iteration of SGDS consists of the following steps:

(1) Generate $m$ new molecules $\{x_i^{t+1}\}_{i=1}^m \sim p_{\theta_t}(x|y = \tilde{y}^t)$. Here, $\tilde{y}^t = y^t + \Delta_y$, where $\Delta_y$ is a small increment, and $y^t$ is the ground-truth property value of a molecule randomly sampled from the current shifting dataset $\mathcal{D}^t$. To accomplish generation, we first sample $z^{t+1} \sim p_{\theta_t}(z|\tilde{y}^t)$ and then use the generation model to get $x^{t+1} \sim p_{\beta_t}(x|z^{t+1})$. To sample $z^{t+1}$, we need to run finite-step Langevin dynamics to sample from $p_{\theta_t}(z_0|\tilde{y}^t)$. This Langevin dynamics is initialized from the corresponding

$z_0$ we keep in the previous shifting iteration, from which we run a very small number of Langevin steps (typically 2 steps).

(2) Annotate the generated molecules using the software (e.g., AutoDock-GPU or RDKit), which is a black-box score or reward function $S(x)$, i.e. $\{y_i^{t+1} = S(x_i^{t+1})\}_{i=1}^m$. Different from [Kong et al., 2023], we do not assume $m = n$. The updated dataset, $\mathcal{G}^{t+1}$, combines the newly generated samples with the previous dataset: $\mathcal{G}^{t+1} = \{x_i^{t+1}, y_i^{t+1}, z_i^{t+1}\}_{i=1}^m \cup \mathcal{D}^t$, amounting to $n + m$ samples. For simplicity in notation, we write $\mathcal{G}^{t+1} = \{x_i^{t+1}, y_i^{t+1}, z_i^{t+1}\}_{i=1}^{n+m}$.

(3) Rank $n + m$ samples based on target property value $y^{t+1}$. This yields $\mathcal{G}^{t+1} = \{x_{(i)}^{t+1}, y_{(i)}^{t+1}, z_{(i)}^{t+1}\}_{i=1}^{n+m}$ where $y_{(1)}^{t+1} \geq y_{(2)}^{t+1} \geq \cdots \geq y_{(m+n)}^{t+1}$. From this, the top-$n$ samples are kept to create a new shifting dataset: $\mathcal{D}^{t+1} = \{x_{(i)}^{t+1}, y_{(i)}^{t+1}, z_{(i)}^{t+1}\}_{i=1}^n$. Additional heuristic constraints can be applied during this selection. For instance, instead of ranking by $y_i^{t+1}$, we might rank by $y_i^{t+1} \mathbb{1}_{S'(x_i)>s}$ where $S'(x_i)$ is another score function, $s$ is the desired threshold and $\mathbb{1}$ is the indicator function.

(4) Update the model parameter $\theta^{t+1}$ by learning from the new shifting dataset $\mathcal{D}^{t+1}$ using Algorithm 2.

The integration of steps (1) and (2) constitutes the primary phase of the SGDS algorithm: dataset creation. Subsequently, the combination of steps (3) and (4) forms the second phase: model shift.

While [Kong et al., 2023] proposes shifting a latent space energy-based model, our aim here is to apply SGDS for shifting our LPT.

## 3 Experiments

We demonstrate our proposed molecule design approach for both single and multi-objective settings.

### 3.1 Experiment Setup

**Dataset.** For molecule property optimization tasks, we use ZINC [Irwin et al., 2012] with 250k molecules. RDKit is used to calculate penalized logP, drug-likeliness (QED) and synthetic accessibility score (SA), and we use docking scores from AutoDock-GPU to approximate the binding affinity to two protein targets, human estrogen receptor (ESR1) and human peroxisomal acetyl-CoA acyl transferase 1 (ACAA1).

**Model Architectures.** As shown in Fig. 1, the prior model is underpinned by Unet1D, assuming 4 latent vectors for $z$ with each sized at 256. The molecule generation model leverages a 3-layer causal Transformer complemented by a cross-attention layer. It has an embedding size of 256 and uses a maximum SELFIES sequence length of 73. The property regression model utilizes a 3-layer MLP, accepting inputs sized at 1024 ($256 \times 4$). The total number of parameters for our LPT is 4.33M.

**Training Details.** We adopt a two-step training approach for LPT. Initially, we pre-train on molecules alone for 30 epochs using Algorithm 1, with a learning rate ranging between $7.5 \times 10^{-4}$ and $7.5 \times 10^{-5}$ via cosine scheduling. Subsequently, we finetune for 10 epochs on both molecules and their properties using Algorithm 2, adjusting the learning rate between $3 \times 10^{-4}$ and $7.5 \times 10^{-5}$. For SGDS process, the total shifting iterations is 25 and the number of new generated samples is set at 2500 for each iteration, with total 62.5k queries of the software. We use the AdamW optimizer [Loshchilov and Hutter, 2019] with 0.1 weight decay for all the above learning processes. Pre-training LPT, fine-tuning LPT and shifting LPT take around 20, 10 and 12 hours respectively on a single NVIDIA A6000.

### 3.2 Binding Affinity Maximization

ESR1 and ACAA1 are human proteins. Our goal is to design ligands with optimal binding affinities to these proteins. While ESR1 has many known binders, SGDS disregards binder-specific data. Binding affinity is measured by the estimated dissociation constants, $K_D$, which can be approximated by AutoDock-GPU's docking scores. A lower $K_D$ indicates stronger binding. Our model excels in the

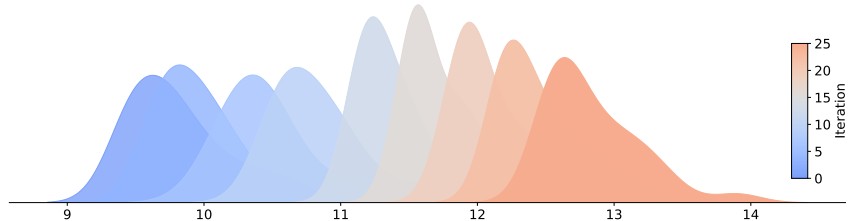

Figure 2: Distribution shift of ACAA1 binding affinity across optimization iterations. For each shift iteration, we plot the densities of property values estimated from AutoDock-GPU.

single objective ESR1 and ACAA1 binding affinity maximization tasks, as highlighted in Table 1. Compared to other state-of-the-arts, it consistently samples high-affinity molecules in the shifting trajectories. Specifically, our latent prompt Transformer outperforms LEBM-SGDS [Kong et al., 2023], showcasing its robust modeling capabilities. Additionally, unlike LEBM, our LPT can readily scale its complexity of prior model and generative Transformer, making it more adaptable to larger datasets and training scenarios.

For multi-objective optimization tasks, we consider maximizing binding affinity, QED and minimizing SA. Meanwhile, we also recruit heuristics to set a threshold to select more probable molecule. In Algorithm 3, we exclude molecules with QED smaller than $0.4$ and SA larger than $5.5$. Results in Table 2 show that LPT is able to get comparable QED and SA to LEBM while getting much higher binding affinities, which demonstrates its superior modeling capability. Generated molecules can be found in Appendix.

Table 1: Single-objective binding affinity optimization. Report top-3 lowest $K_D$ (in nanomoles/liter) found by each model. Baseline results obtained from [Eckmann et al., 2022, Kong et al., 2023].

| Method | ESR1 $K_D$ ($\downarrow$) | | | ACAA1 $K_D$ ($\downarrow$) | | |
|---|---|---|---|---|---|---|
| | 1st | 2rd | 3rd | 1st | 2rd | 3rd |
| GCPN | 6.4 | 6.6 | 8.5 | 75 | 83 | 84 |
| MolDQN | 373 | 588 | 1062 | 240 | 337 | 608 |
| MARS | 25 | 47 | 51 | 370 | 520 | 590 |
| GraphDF | 17 | 64 | 69 | 163 | 203 | 236 |
| LIMO | 0.72 | 0.89 | 1.4 | 37 | 37 | 41 |
| LEBM-SGDS | 0.03 | 0.03 | 0.04 | 0.11 | 0.11 | 0.12 |
| **LPT-SGDS** | **0.004** | **0.005** | **0.014** | **0.037** | **0.046** | **0.084** |

Table 2: Muli-objective optimization for both ESR1 and ACAA1. Report Top-2 average scores of $K_D$ (in nmol/L), QED and SA. Baseline results obtained from [Eckmann et al., 2022, Kong et al., 2023].

| Ligand | ESR1 | | | ACAA1 | | |
|---|---|---|---|---|---|---|
| | $K_D \downarrow$ | QED $\uparrow$ | SA $\downarrow$ | $K_D \downarrow$ | QED $\uparrow$ | SA $\downarrow$ |
| Tamoxifen | 87 | 0.45 | 2.0 | – | – | – |
| Raloxifene | $7.9 \times 10^6$ | 0.32 | 2.4 | – | – | – |
| GCPN 1st | 810 | 0.43 | 4.2 | 8500 | 0.69 | 4.2 |
| GCPN 2nd | 27000 | 0.80 | 3.7 | 8500 | 0.54 | 4.3 |
| LIMO 1st | 4.6 | 0.43 | 4.8 | 28 | 0.57 | 5.5 |
| LIMO 2nd | 2.8 | 0.64 | 4.9 | 31 | 0.44 | 4.9 |
| LEBM-SGDS 1st | 0.36 | 0.44 | 3.99 | 4.55 | 0.56 | 4.07 |
| LEBM-SGDS 2nd | 1.28 | 0.44 | 3.86 | 5.67 | 0.60 | 4.58 |
| **LPT-SGDS** 1st | **0.05** | 0.46 | **3.24** | **0.06** | 0.57 | 4.54 |
| **LPT-SGDS** 2nd | **0.05** | 0.60 | 5.02 | 0.08 | 0.48 | **4.01** |

# 4  Related Work

Our model is based on [Kong et al., 2023]. The difference are as follows. (1) While [Kong et al., 2023] used the LSTM model for molecule generation, we adopt a more expressive causal Transformer model for generation, with the latent vector serving as latent prompt. (2) While [Kong et al., 2023] used a latent space energy-based model for the prior of the latent vector, we assume that the latent $z$ is generated by a Unet transformation of a Gaussian white noise vector. This enables us to avoid the Langevin dynamics for prior sampling in learning, thus simplifies the learning algorithm. (3) We obtain much stronger experimental results, surpassing [Kong et al., 2023] and achieving new state of the art performances.

Compared to existing latent space generative models [Gómez-Bombarelli et al., 2018, Kusner et al., 2017, Jin et al., 2018, Eckmann et al., 2022], we assume a learnable prior model so that our model can adeptly catch up with the shifting dataset in the optimization process.

Compared to population-based methods such as genetic algorithms [Nigam et al., 2020] and particle-swarm algorithms [Winter et al., 2019], our method does not only maintain a shifting dataset (which can be considered a small population), but also a shifting model to fit the dataset, so that we can generate new molecules from the model. The model itself is virtually an infinite population because it can generate infinitely many new samples.

# 5  Conclusion

This paper proposes a latent prompt Transformer model for molecule design. We assume the solution can be represented by a sequence of tokens. We employ a latent prompt that generates the sequence via a causal Transformer model and predicts the value of the target property via a regression model. We develop the approximate maximum likelihood learning algorithm and we employ the gradual distribution shifting algorithm for optimization with learning in the loop. Our proposed method achieves new state of the art on several benchmark tasks on molecule design.

Our model and method can be applied to on-line black-box optimization problem in general, and the Transformer model can be replaced by other conditional generation models if the solution is not in the form of a sequence of tokens. In our future work, we shall explore applying our method to other challenging optimization problems in science and engineering.

# Acknowledgement

Y. N. Wu was partially supported by NSF DMS-2015577 and a gift fund from Amazon.

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

## Appendix

We display molecules generated by LPT as they evolve throughout the shifting trajectories.

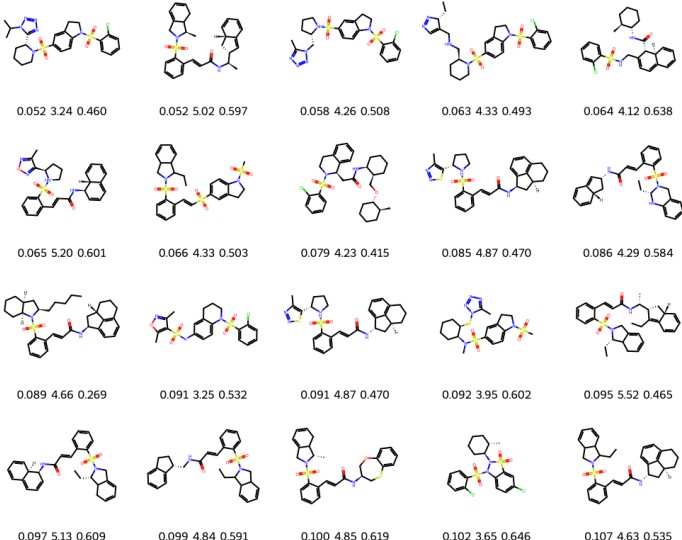

Figure 3: Molecules produced during the multi-objective optimization for ESR1. The legends denote $K_D \downarrow$, SA$\downarrow$ and QED$\uparrow$.

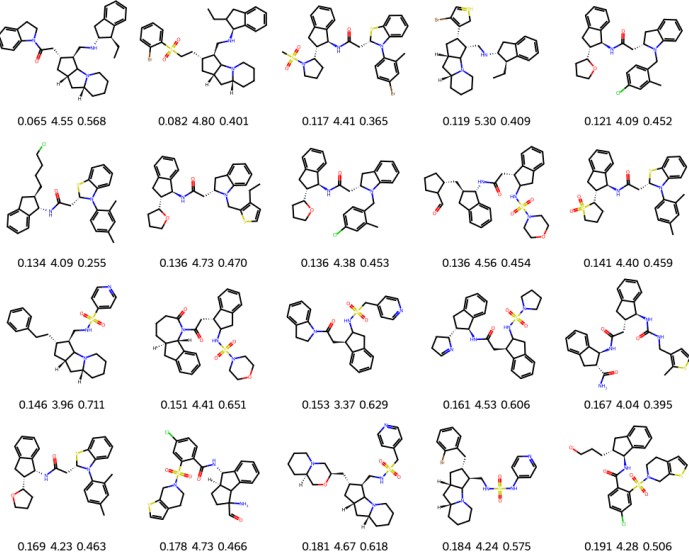

Figure 4: Molecules produced during the multi-objective optimization for ACAA1. The legends denote $K_D \downarrow$, SA$\downarrow$ and QED$\uparrow$.

