# OpenReview forum: "Molecule Design by Latent Prompt Transformer"
_NeurIPS.cc/2023/Workshop/AI4Science — NeurIPS2023-AI4Science Poster_

### Official Review · Reviewer_kfvk · 2023-10-16
**The article proposes a novel latent prompt Transformer model for molecule design, which consists of three components: a latent vector, a molecule generation model, and a property prediction model.**

**Rating:** 6
**Confidence:** 3

**Review:**

The article proposes a  novel latent prompt Transformer model for molecule design. The model consists of three components: a latent vector, a molecule generation model, and a property prediction model. The latent vector is generated by a Unet transformation of a Gaussian white noise vector. The molecule generation model uses a causal Transformer model with the latent vector as a prompt. The property prediction model uses a non-linear regression on the latent vector. The model is trained using an approximate maximum likelihood learning algorithm and a gradual distribution shifting algorithm. Experimental results demonstrate that the proposed model achieves  state-of-the-art performance  on molecule design tasks. The methodology is appropriate and rigorous. The paper is well-written and easy to understand.

---

### Official Review · Reviewer_nrts · 2023-10-25
**Promising new method but more evaluation is needed**

**Rating:** 6
**Confidence:** 4

**Review:**

Summary: This paper develops a generative model for molecule design. The authors develop a three component model that includes a U-Net to map Gaussian white noise to a latent vector, a molecule generation model to output a molecular string based on the latent vector, and a property prediction model to predict the property of the molecule from its latent vector. The authors design a training scheme to first train the model on unlabeled molecules followed by conditional generation with a distribution shifting algorithm to slowly move the model toward parts of the latent space containing high-scoring molecules. The authors show that their model outperforms existing models when designing compounds for two protein targets.

Strengths: The paper designs a theoretically motivated generative model that breaks down the generative process into three reasonable steps. The authors design a training scheme that leverages unlabeled data and overcomes the issue of out-of-distribution generation by slowly shifting the distribution of the generated molecules. These design choices make sense and lead to strong results in terms of top molecule docking scores for two proteins.

Weaknesses: Although the top two or three generated molecules have impressive docking scores, more analysis is needed to further evaluate the proposed generative model. It would be valuable to report not just the top molecule docking scores but the entire distribution of docking scores across the generated molecules (or at least the mean and standard deviation). This is important because even though the goal of drug discovery might be to find a single drug, there may be many other druglike properties that the top scoring generated candidate fails to possess, so it would is useful to have a generative model that designs many good candidates, not just one or two.

Along those lines, it would be valuable to measure the chemical diversity of the generated molecules. A visual inspection of Figures 3 and 4 shows several motifs that are repeated across many molecules, so the diversity of the generated molecules may be less than the diversity of molecules generated by other methods. Again, diversity is important to provide multiple options for medicinal chemists since the molecule with the best docking score may not be suitable as a drug for many reasons.

The authors are also encouraged to report the QED and SA scores across all the generated molecules. The QED and SA scores of the top molecules are not particularly impressive compared to the other baseline methods, so it would be valuable to know if this is true across the rest of the generated molecules.

It is also important to note how many molecules each of the baseline methods generated. If the baseline methods generated fewer (or more) molecules than the proposed approach, then the comparison of top molecules would not be fair since certain methods may have more opportunities to generate a top scoring molecule than others.

Finally, the authors should clarify the scale used on the x-axis of Figure 2. How do these values relate to K_D?

---

### Meta-Review · Area_Chair_Wf8m · 2023-10-27

**Recommendation:** Accept (Poster)
**Confidence:** 5

**Metareview:**

This paper proposes latent prompt Transformer model, a latent diffusion-like model for molecule design. One limitation of this work is the lack of discussion on existing molecule conditional generation / molecule editing / lead optimization work using deep learning. Yet, this is still an interesting paper that can spark more insights in this field.